# COVID-19 Vaccine: Between Myth and Truth

**DOI:** 10.3390/vaccines10030349

**Published:** 2022-02-23

**Authors:** Pier Paolo Piccaluga, Antonio Di Guardo, Anna Lagni, Virginia Lotti, Erica Diani, Mohsen Navari, Davide Gibellini

**Affiliations:** 1Department of Experimental, Diagnostic, and Specialty Medicine, Institute of Hematology and Medical Oncology “L. and A. Seràgnoli”, University of Bologna School of Medicine, 40126 Bologna, Italy; antonio.diguardo@studio.unibo.it; 2SBGT—Biomolecular Strategies, Genetics and Cutting-Edge Therapies, Istituto Euro-Mediterraneo di Scienza e Tecnologia (IEMEST), 90139 Palermo, Italy; 3Department of Pathology, School of Medicine, Jomo Kenyatta University of Agriculture and Technology, Juja 01001, Kenya; 4School of Medicine, Nanchang University, Nanchang 330047, China; 5Department of Diagnostic and Public Health, Verona University, 37134 Verona, Italy; anna.lagni@univr.it (A.L.); virginia.lotti@univr.it (V.L.); erica.diani@univr.it (E.D.); davide.gibellini@univr.it (D.G.); 6Department of Medical Biotechnology, School of Paramedical Sciences, Torbat Heydariyeh University of Medical Sciences, Torbat Heydariyeh 33787 95169, Iran; mohsen.navari@gmail.com; 7Research Center of Advanced Technologies in Medicine, Torbat Heydariyeh University of Medical Sciences, Torbat Heydariyeh 33787 95169, Iran; 8Bioinformatics Research Group, Mashhad University of Medical Sciences, Mashhad 91778 99191, Iran

**Keywords:** SARS-CoV-2, coronavirus, COVID-19, vaccine, immunization, viral vector, mRNA

## Abstract

Since December 2019, a pandemic caused by the newly identified SARS-CoV-2 spread across the entire globe, causing 364,191,494 confirmed cases of COVID-19 to date. SARS-CoV-2 is a betacoronavirus, a positive-sense, single-stranded RNA virus with four structural proteins: spike (S), envelope (E), membrane (M), and nucleocapsid (N). The S protein plays a crucial role both in cell binding and in the induction of a strong immune response during COVID-19 infection. The clinical impact of SARS-CoV-2 and its spread led to the urgent need for vaccine development to prevent viral transmission and to reduce the morbidity and mortality associated with the disease. Multiple platforms have been involved in the rapid development of vaccine candidates, with the S protein representing a major target because it can stimulate the immune system, yielding neutralizing antibodies (NAbs), blocking viral entry into host cells, and evoking T-cell immune responses. To date, 178 SARS-CoV-2 vaccine candidates have been challenged in clinical trials, of which 33 were approved by various national regulatory agencies. In this review, we discuss the FDA- and/or EMA-authorized vaccines that are mostly based on mRNA or viral vector platforms. Furthermore, we debunk false myths about the COVID-19 vaccine as well as discuss the impact of viral variants and the possible future developments.

## 1. Introduction

In December 2019, a cluster of acute pneumonia cases with unknown etiology was detected in Wuhan, China. Afterwards, a new human coronavirus, the Severe Acute Respiratory Syndrome Coronavirus 2 (SARS-CoV-2), was determined as the cause of this disease and was named coronavirus disease 2019 (COVID-19) [1]. This virus spread rapidly across the world. The World Health Organization (WHO) declared a pandemic on March 11, 2020, and, according to their latest report [2], there are currently 364,191,494 confirmed cases of COVID-19 and 5,631,457 deaths (current as of 28 January 2022).

SARS-CoV-2 is mainly transmitted through respiratory secretions, and infected patients can show a broad spectrum of clinical manifestations, ranging from asymptomatic to critical illness with acute respiratory failure, septic shock, multiorgan dysfunction, and in the worst cases, death. Fever, headache, cough, fatigue, anosmia or dysgeusia, sore throat, myalgia, and breathlessness are common clinical characteristics of this emerging disease [3].

SARS-CoV-2 is a member of the subgenus Sarbecovirus (betacoronavirus lineage B) of the *Coronaviridae* family. Coronaviruses (CoVs) are cytoplasmic replicating, positive-sense, single-stranded RNA viruses with four structural proteins: spike (S), envelope (E), membrane (M), and nucleocapsid (N). The S protein plays a crucial role in cell binding and in eliciting the immune response during the progression of the disease [4]. SARS-CoV-2 enters host cells via the binding of the S protein with the angiotensin-converting enzyme 2 (ACE2) receptor. Cell entry is also regulated by coreceptors such as transmembrane serine protease 2 (TMPRSS2) [5,6,7] and furin, which are both responsible for S protein cleavage and the priming [8] necessary for SARS-CoV-2 cell entry [9,10].

In the last two decades, emerging coronaviruses have been observed to be highly pathologic to humans. Before December 2019, only six members of the *Coronaviridae* family (229E, NL63, OC43, HKU1, MERS-CoV, and SARS-CoV) had been isolated and characterized as causes of respiratory human disease. Outbreaks of SARS-CoV (2003) and MERS-CoV (2009) caused severe respiratory diseases with a mortality rate of 10% and 36%, respectively, even with a limited number of infected patients [11]. The clinical impact of SARS-CoV-2 and its spread was different from the previously known coronaviruses; it led to the urgent need for the development of a vaccine to limit transmission and to reduce the morbidity and mortality associated with the disease, and multiple platforms have been involved in the rapid development of vaccine candidates. The major target for the vaccine development is the S protein because the immune system is able to yield neutralizing antibodies (NAbs) against this protein, block viral entry into host cells, and induce T-cell immune responses [12]. To date (26 January 2022), 178 SARS-CoV-2 vaccine candidates have been challenged in clinical trials and 33 vaccine candidates have been approved by various national regulatory agencies. To achieve this goal, multiple platforms and technologies were used in the rapid development of SARS-CoV-2 vaccine candidates, using both classical and new strategies, including mRNA, adenoviral vector-based, adjuvanted recombinant protein, and inactivated virus vaccines [13].

As of 28 January 2022, a total of 9,854,237,363 vaccine doses have been administered [2], and the most widely used are mRNA vaccines, including the BNT162b2 (Pfizer-BioNTech, New York, NY, USA—Mainz, Germany) and mRNA-1273 (Moderna, Cambridge, MA, USA) vaccines, and viral vector vaccines, such as Ad26.CoV2.S (Johnson & Johnson, New Brunswick, NJ, USA), ChAdOx (AstraZeneca, Cambridge, UK), Sputnik V (Gamaleya Research Institute of Epidemiology and Microbiology, Moscow, Russia) and the inactivated virus alum-adjuvanted candidate vaccine CoronaVac (Sinovac, Beijing, China). All of these vaccines were well tolerated in clinical trials and their proven efficacy was greater than 90% in preventing symptomatic disease, except for the CoronaVac vaccine, which had a proven effectiveness of 51% [14,15].

In this review, we will focus our attention on the false information and truths regarding mRNA and viral vector vaccines authorized by the Food and Drug Administration (FDA) and the European Medical Agency (EMA) (Table 1). We will also consider variants of concern (VOCs), implications on their efficacy, and possible future developments.

## 2. COVID-19 Vaccines: Myths and Hesitancy

Once the main hurdle of finding a way to tackle the COVID-19 pandemic was overcome, other challenges arose in the COVID-19 immunization campaign: skepticism and fear of vaccinations [16]. This phenomenon has been named by the WHO as “vaccine hesitancy” and is defined as “the reluctance or refusal to vaccinate despite the availability of vaccines” [17]. Hesitancy towards vaccinations is a phenomenon that has existed since the first vaccine became available. Vaccine hesitancy is a complex problem closely linked to the individuals social context, income, and level of education, posing dangers to both the individual and their community and increasing the risk of vaccine-preventable disease outbreaks and epidemics [18,19].

In particular, the anti-vaccine community argues that vaccines, contrary to the claims of most of the scientific community do not have a sufficient or adequate safety profile. Major theories against vaccination include the widespread fear that vaccines increase the risk of developing autism, a theory that originated from a study published in 1997 in *The Lancet* by Andrew Wakefield; in this study, the authors suggested that the measles, mumps, and rubella (MMR) vaccine caused an increasing incidence of autism in British children. The paper has since been discredited due to serious procedural errors, undeclared financial conflicts of interest, and ethical violations [20]. Another recurrent reason for vaccine refusal is the presumed high number of dangerous substances in vaccine formulations, such as formaldehyde, mercury, and aluminum. There is no scientific basis to confirm these claims, and it has been reported that the low levels of mercury or aluminum in vaccines cannot be harmful and, moreover, that formaldehyde is produced at higher rates by our own metabolic systems [21].

Another theory accepted by vaccine skeptics is that natural immunity (the immunity that develops after infection) is superior to vaccine-acquired immunity. In some cases, natural immunity results in a stronger immunity to the disease than that provided by vaccination; however, the dangers of this approach far outweigh the relative benefits [22], as in the case of COVID-19 whereby the prevalence of 55 long-term effects was evaluated [23].

Among the conspiracy theories of the anti-vaccine movement, a key role is played by the observation that vaccines are one of the main sources of income for pharmaceutical companies (often given the nickname “Big Pharma”), and therefore the establishment of a major vaccine plan with multiple mandatory vaccinations would do nothing but increase profits at the expense of the health care system, and not the provision of greater patient care and disease prevention, as, in fact, is the case.

In the current day, the uncertainty caused by the pandemic and controversies surrounding vaccine safety that circulate in news headlines, talk shows, and popular articles, in addition to the spread of “fake news” on social media, have made vaccine hesitancy one of the main themes in the fight against COVID-19 [16]. This spread of misinformation has been defined by the WHO as another epidemic to be fought, called lymphodemics [18].

Regarding COVID-19 vaccines specifically, one of the first and major concerns in the collective ideal and around the anti-vaccine community was their rapid development and release, making it difficult to trust their safety and effectiveness. For this reason, many efforts were made by the scientific community to disclose the factors responsible for the record time of COVID-19 vaccine release: first, past research on mRNA technology and on human coronaviruses related to SARS-CoV-2 (such as SARS-CoV and MERS-CoV) were key starting points in the research of vaccines against COVID-19, allowing for the reduction in development time; because the development occurred in the midst of a pandemic, substantial human and economic resources were made available under tight time constraints, permitting research to be conducted in parallel with the various phases of evaluation and study; as the pandemic was a global emergency, evaluation of the results by regulatory agencies was made straight-forward after the completion of each phase (rolling review), and not at the completion of all studies, as is generally the case [21,24].

Another main concern of anti-vaccine community about the vaccines is the belief in their ability to alter human DNA with mRNA technology, due to misinformation that led to ignorance of the cornerstones on which biology and medicine are founded. The mRNA contained in COVID-19 vaccines never enters the cell nucleus; moreover, it is sensitive to degradation by ubiquitous ribonucleases (RNase) with metabolic decay occurring within a few days. Regarding the viral vector vaccines, the concern was the same as with mRNA, but studies reported that, even in this case, the genetic material delivered into host cells reached the nucleus without integrating into the host DNA [25,26].

Recently, there has been much debate about whether variants may be caused by COVID-19 vaccine administration, as the anti-vaccine community claims that, to escape a vaccinated host’s immune system, the virus should be able to modify its genetic material, which leads to development of variants. However, it is important to note that the main issue leading to variant development concerns virus circulation and consequently the non-immunized population, representing an ideal environment for the development of mutations due to the absence of a defense, allowing the virus to replicate undisturbed. The more mutations that occur, the more likely it is that variants will appear. Vaccines are able to decrease the viral load in case of infection, thereby reducing the number of reproduction cycles of the virus by decreasing the possibility of new mutations and the emergence of new variants [27,28].

Given the hesitancy and false information surrounding SARS-CoV-2 vaccines, the key responsibilities of policy, health systems, and healthcare professionals include the need to clear any doubts about vaccine safety and to promote an effective, science-based information campaign.

## 3. SARS-CoV-2 mRNA-Based Vaccines

To date, the mRNA-based vaccine platform represents a promising alternative to conventional vaccines and provides the most rapidly accessible vaccine candidates due to their short development time, scalability, low-level biosafety requirement, safe administration, and ability to generate Th1 and Th2 responses [29]. Although this is a novel technology, it has already been used to develop vaccines against other infectious diseases, such as the influenza virus, respiratory syncytial virus (RSV), Zika virus, Ebola virus, HIV, and also against several types of cancer [26].

The delivery of mRNA in non-toxic carriers was the main challenge to ensure its efficacy because mRNA is an unstable and easily degraded molecule, particularly due to the widespread presence of RNases in the human body. Thanks to recent technological advantages, novel vaccine designs were developed to improve the stability and efficiency of protein translation, to enhance the immune response, in particular the use of lipid nanoparticle (LNP)-encapsulated mRNA that facilitates the release of mRNA into cells from endosomes [30].

An LNP consists of a bilayer lipid vesicle with a surface coating of polyethylene glycol (PEG), and cholesterol, phospholipids, and ionizable lipids in the structure. This central component is essential for the release of mRNA molecules after intramuscular (IM) injection, with a neutral charge at physiological pH and a protonated form at a lower pH in the endosome [31]. When injected intramuscularly, the adjuvant effect of mRNA-LNPs allows mRNA to be internalized and quickly translated by antigen-presenting cells (APCs) at both the injection site and in draining lymph nodes, promoting the initiation of the adaptive immune response [30]. In a study assessing an ionizable LNP formulation, an adjuvant effect was observed from the ionizable lipid component and IL-6 cytokine induction that elicited strong T follicular helper cell, germinal center B-cell, long-lived plasma cell, and memory B-cell responses that were associated with the production of long-lasting and protective antibodies in mice [32].

Once the mRNA vaccine is injected into the patient, the lipid nanoparticles protect the mRNA from degradation and promote its entry into cells where the mRNA is then translated to produce the viral protein to elicit a specific immune response. The produced viral proteins stimulate the immune system primarily through T-cells and the synthesis of neutralizing antibodies, which aim to prevent SARS-CoV-2 infection (Figure 1) [33]. In vaccinated patients subsequently exposed to the virus, the antibodies block the S protein and prevent viral entry into the cells. mRNA is highly immunogenic with a self-adjuvating effect, activating several pathogen-associated molecular pattern sensors; however, the early triggering of a type I interferon (IFN-1) response can downregulate protein expression [34]. Therefore, in order to avoid excess inflammation and to improve the half-life and safety, modified nucleosides can incorporate into the mRNA to create a “silenced” RNA vaccine that avoids activation of the toll-like receptor (TLR) mechanism or the retinoic-acid-inducible gene and does not trigger an IFN-1 response [35]. In addition, these vaccines elicit adaptive cellular responses, stimulating Thelper cell and cytotoxic T lymphocytes (CTLs) that play a pivotal role in the immune response. Some studies have reported a rapid decline in anti-SARS-CoV-2 IgG levels in mild COVID-19 patients [36]. mRNA vaccines are safe because they are non-infectious as they are not made with pathogen particles or inactivated pathogens, and they are effective as the antigen is expressed in vivo and induces both humoral and cell-mediated immune responses. Moreover, the manufacturing of nucleic acid vaccines can be synthetic and entirely cell-free, avoiding the need for BSL2 laboratories; furthermore, the production is rapid and the process can be standardized, improving the responsiveness to emerging outbreaks [37]. However, due to the low stability and uncertainty surrounding the formulation of mRNA vaccines, no mRNA vaccine candidates were commercialized before this pandemic; nonetheless, with recent technical advances, several institutions around the world have begun rapid development of mRNA and DNA vaccines [38]. The Pfizer-BioNTech (BNT162b2) and Moderna (mRNA-1273) vaccines were the first vaccines to reach Phase IV and involve two-dose regimens of the vaccine formulations, with 2 to 3 weeks between each IM dose. High efficacy was reported with these vaccines (more than 94% efficacy in Phase III clinical trials) and they triggered a rapid and effective immune response, making them the perfect candidates as their mechanisms utilized the capacity of the host cells to translate the encoding mRNA to SARS-CoV-2 S proteins [39].

### 3.1. Comirnaty (BNT162b2)—Pfizer-BioNTech Vaccine

The Pfizer-BioNTech COVID-19 vaccine was the first vaccine approved in the United States and it has been available since 11 December 2020, under EUA (Emergency Use Authorization) for people 16 years and older. The authorization was expanded to include people aged 12 to 15 years on 10 May 2021, and children aged 5 to 11 years on 29 October 2021 [40]. It was definitively approved on 23 August 2021, for the prevention of COVID-19 disease in individuals aged 16 years and older, and is now marketed as Comirnaty [41]. The EMA granted conditional marketing authorization for this vaccine in people 18 years and older on 21 December 2020 [42]. On 28 May 2021, the EMA recommended granting an extension of indication for the Comirnaty vaccine to be used in children aged 12 to 15 years [43], and on 25 November 2021 it recommended the vaccines for use in children aged 5 to 11 years [44].

BNT162b2 encodes a membrane-anchored, full-length SARS-CoV-2 spike protein modified by two proline mutations to stabilize the protein in its prefusion conformation [45]. Once thawed, the vaccine can be stored for up to 5 days before use at a temperature between 2 and 8 °C, and up to 2 h at a temperature not exceeding 30 °C. Before use, BNT162b2 must be reconstituted with 0.9% sodium chloride. Once reconstituted, it should be administrated within 6 h and stored between 2 and 30 °C. Comirnaty is administered as two IM doses injected 21 days apart, and each vial contains six doses [46].

BNT162b2 was selected for phase 2–3 of safety and efficacy evaluation due to its low reactogenicity [45], a significant antibody response, and IFN-γ or IL-2 CD8^+^ and CD4^+^ T helper type 1 (Th1) cell responses [47].

The efficacy of BNT162b2 was evaluated in approximately 44,000 people; based on results from the clinical trial (NCT04368728), the vaccine was 94.6% effective in preventing COVID-19 disease 7 days after the second dose (95% CI, 89.9–97.3%) [42,48].

Regarding the safety profile, a higher frequency of local reactions including mild-to-moderate pain at the injection site and less frequent redness or swelling was observed in patients who received this vaccine. Systemic reactogenicity was reported more frequently in younger vaccine recipients and more frequently after the second dose. The most common systemic adverse events were fatigue and headache [49].

Data on the immunological response [50] reported that vaccination with BNT162b2 induced a coordinated immune response with SARS-CoV-2 S-protein-specific neutralizing antibodies, CD4^+^ T-cells, CD8^+^ T-cells, and immune-modulatory cytokines such as IFN-γ. In particular, it was observed that there was a de novo S-specific CD4^+^ T-cell response in all vaccinated participants, and almost 92% of participants mounted CD8^+^ T-cell responses; furthermore, even with the lowest dose of BNT162b2, most of the vaccinated participants demonstrated robust expansion of CD4^+^ and CD8^+^ T-cells. This response was directed against the receptor-binding domain (RBD) S1 and S2 regions of the S protein, indicating immune recognition of multiple independent major histocompatibility complex (MHC) I and II epitopes. Plasma IgM, IgG, and IgA responses to SARS-CoV-2 S regions and the RBD were measured by enzyme-linked immunosorbent assay (ELISA). All tested individuals showed significant reactivity to S and RBD and, as expected, levels of anti-S and anti-RBD IgG were higher than levels of IgM or IgA. Moreover, there was a strong positive correlation between the anti-RBD and anti-S responses for all three immunoglobulin isotypes. It is not yet known if the antibody responses will be sufficient for full and long-lasting protective immunity [51], but with large-scale immunization in Israel (the first nation to begin vaccination), the first data on the effectiveness of the BNT162b2 vaccine in a real-world setting were reported. Adjusted estimates of vaccine effectiveness at 7 days or longer after the second dose were 95.3% (incidence rate 91.5 per 100,000 person-days in unvaccinated vs. 3.1 per 100,000 person-days in fully vaccinated individuals) against SARS-CoV-2 infection, 91.5% (40.9 vs. 1.8 per 100,000 person-days) against asymptomatic SARS-CoV-2 infection, 97% (32.5 vs. 0.8 per 100,000 person-days) against symptomatic COVID-19, 97.2% (4.6 vs. 0.3 per 100,000 person-days) against COVID-19-related hospitalization, 97.5% (2.7 vs. 0.2 per 100,000 person-days) against severe or critical COVID-19-related hospitalization, and 96.7% (0.6 vs. 0.1 per 100,000 person-days) against COVID-19-related death. In all age groups, as vaccine coverage increased, the incidence of SARS-CoV-2 outcomes declined [52].

### 3.2. Spikevax (mRNA-1273)—Moderna Vaccine

On 18 December 2020, the US FDA issued an EUA for use of the Moderna vaccine in individuals 18 years of age and older to prevent COVID-19. The EMA recommended granting a conditional marketing authorization for the COVID-19 Moderna vaccine to prevent COVID-19 in people 18 years of age and older on 6 January 2021 [53,54], and, on 23 July 2021, this vaccine was authorized for use in children aged 12 to 17 years [55].

mRNA-1273 is an mRNA vaccine encoding the SARS-CoV-2 S-2P antigen, stabilized in its prefusion conformation by two consecutive proline substitutions at positions 986 and 987 [56]. The administration of the mRNA-1273 vaccine includes two 0.5 mL doses injected intramuscularly 28 days apart. The vaccine can be stored between −25 and −15 °C for up to 7 months, but can be stored between 2 and 8 °C for 30 days. No dilution or reconstitution is required. Before the administration, doses can be maintained at room temperature (8–25 °C) for up to 12 h. Once the vial is opened, it should be stored at between 2 and 25 °C for no more than 6 h. Each vial contains ten doses [57].

This vaccine demonstrated 94.1% efficacy (95% CI, 89.3–96.8%) in a trial (NCT04470427) involving around 30,000 people. The most common local adverse reactions were pain at the injection site and, more rarely, erythema, induration, and tenderness. Frequent systemic side effects were fatigue, headache, myalgia, arthralgia, chills, nausea/vomiting, lymphadenopathy in the injection arm, and fever, and these side effects were mostly mild and transient. Reactions were more frequently reported after the second dose and in younger recipients [39]. The Vaccine Adverse Event Reporting System (VAERS) has identified rare cases of myocarditis and pericarditis following administration of Spikevax. Myocarditis and pericarditis occurred mainly in the first week following vaccination, more often after the second dose, and more frequently in young males [58].

Like BNT162b2, this vaccine elicits an IgG immune response together with cellular responses and is biased mainly towards CD4^+^ Th1 cells, while the CD8^+^ T-cell responses were marginal except for those in recipients of two vaccinations with the higher dose [59].

## 4. Viral Vector Vaccines

Viral vector vaccines use replication-deficient viruses (usually a modified adenovirus) genetically engineered to express the genetic sequence of the antigen of interest in host cells. Some of the vectors—like many of the adenoviral vectors—are developed to be replication incompetent. Adenovirus vectors are non-enveloped double-stranded DNA (dsDNA) viruses with a packaging capacity of 7.5 kb of foreign DNA, providing transient episomal expression in a broad range of host cells [60]. These vectors contain deletions in the native viruses that make them replication incompetent for safety; in particular, E1A and E1B encoding regions, which are needed for replication in infected cells, are deleted and replaced with the target gene [61]. Replication-incompetent adenovirus vectors have been developed with variable success for HIV, tuberculosis, malaria, and Ebola virus, often limited in efficacy due to pre-existing immunity to the adenovirus vector [62]. The main advantage of vector vaccines is that the immunogenic antigen is produced within a heterologous viral infection, leading to the innate immune responses necessary to induce the adaptive immune response, including a robust cytotoxic T lymphocyte (CTL) response. This response subsequently leads to the elimination of virus-infected cells without the need for an adjuvant. Additionally, the long duration and high level of antigenic protein expression and its rapid production makes recombinant viral vectors a popular platform for vaccine delivery. However, this strategy may induce prior immunity to the vector and is limited to presenting only a small number of SARS-CoV-2 antigens to the host immune system [63].

Some of the non-replicating SARS-CoV-2 viral vector vaccines were based on adenovirus types 5 and 26 (Ad5, Ad26) expressing the S protein or RBD subunit of SARS-CoV-2; however, to bypass the action of anti-adenoviral antibodies that may already be present in the recipients, an alternative chimpanzee adenovirus vector (ChAd) with low human prevalence was employed as a vaccine platform [60].

Vaccines that have been granted EUA by the EMA and/or FDA are adenovirus serotype 26 vector vaccine Ad26.CoV2.S (Johnson & Johnson, New Brunswick, NJ, USA) and chimpanzee adenovirus vector vaccine ChAdOx (AstraZeneca, Cambridge, UK). Both vaccines have proven efficacy in preventing hospitalization and death but have varying efficacy in preventing clinical disease [62].

### 4.1. Vaxzevria (ChAdOx1-S)—AstraZeneca/Oxford Vaccine

In January 2021, the EMA granted use authorization of the ChAdOx1-S vaccine, developed by the University of Oxford and AstraZeneca for the prevention of COVID-19 disease in people aged 18 years and older [64]. In March 2021, the EMA approved Vaxzevria as the third vaccine after Comirnaty and Spikevax [65]. The AstraZeneca vaccine was never authorized for use in the US [66].

Vaxzevria is a viral vector vaccine that exploits a different approach to inducing an immune response than the Pfizer-BioNTech and Moderna’s vaccines. ChAdOx1-S consists of a modified version of the chimpanzee adenovirus that is no longer able to replicate, and contains the full-length structural surface S glycoprotein of SARS-CoV-2 with a tissue plasminogen activator leader sequence. The viral vector was obtained from genetically modified human embryonic kidney (HEK) 293 cells and through recombinant DNA technology [67].

The vaccination course with Vaxzervria consists of two separate doses of 0.5 mL each, administered intramuscularly. The second dose should be given 4 to 12 weeks (28 to 84 days) after the first dose [65].

The Oxford COVID-19 Vaccine Trial Group conducted a phase 1–2 trial to investigate the immunogenicity, reactogenicity, and safety of vaccination with ChAdOx1-S [67], and reported local and systemic reactogenicity, such as injection site pain, myalgia, headache, asthenia, chills, and fever. No severe adverse events related to ChAdOx1-S administration were observed. The study showed the induction of a humoral response, characterized by anti-spike glycoprotein IgG and neutralizing antibodies, and IFN-γ T-cell responses in most recipients after the first dose of vaccine and an additional increase in humoral immune response after the second dose of vaccine. The neutralizing antibody response was stronger in the booster vaccination group, and the vaccine elicited a strong T-cell response in all subjects [67]. Subsequently, the ChAdOx1-S candidate vaccine was evaluated in terms of safety and efficacy in four phase 2–3 clinical trials, carried out in the UK, Brazil, and South Africa. Between April and November 2020, 23,848 people over the age of 18 years were selected. This vaccine showed an efficacy of 64.1% against severe COVID-19 after one dose (95% CI, 50.5–73.9%) and 70.4% 14 days after the second dose (95% CI, 54.8–80.6%) [68,69].

Rare side effects that may have a causal relationship with Vaxzevria injection have been reported since the start of the large-scale immunization campaign. These adverse events were collected by the EMA monitoring system and published in safety updates [65]. Guillain–Barré syndrome (GBS) and a thrombotic syndrome associated with thrombocytopenia, sometimes accompanied by hemorrhages, were rarely observed after vaccination with Vaxzevria. Thrombotic syndrome with thrombocytopenia and coagulation disorders may include severe cases of venous thrombosis in atypical sites such as cerebral venous sinus thrombosis, splanchnic venous thrombosis, and arterial thrombosis. Most coagulation disorders occurred in the first 3 weeks after vaccination and mainly in women under the age of 60 years. Due to the increased risk of thromboembolic adverse events, especially in young and female individuals, several countries have recommended the use of the Vaxzevria vaccine in elderly people. For example, in Italy, the Ministry of Health recommends preferential use of the Vaxzevria vaccine in people over the age of 60 years [70]. People younger than 60 years of age who have already received their first dose of AstraZeneca vaccine may complete the vaccine course with the same vaccine or may receive an mRNA vaccine as a second dose after 8–12 weeks [70].

Insufficient data are available to establish the safety and efficacy of Vaxzevria in children and adolescents (individuals less than 18 years old) [65].

### 4.2. Janssen COVID-19 Vaccine (Ad26.COV2.S)—Janssen Pharmaceutical and Johnson & Johnson

In February 2021, the FDA issued an EUA for a single-dose COVID-19 vaccine developed by the Janssen Pharmaceutical Companies of Johnson & Johnson to prevent COVID-19 in people aged 18 years and older [71]. In March 2021, the vaccine received conditional marketing authorization from the EMA [72].

The Ad26.COV2.S is a recombinant, replication-incompetent human adenovirus type 26 vector encoding a full-length SARS-CoV-2 S protein in a prefusion-stabilized conformation [73].

Ad26.COV2.S can be stored for up to 2 years in a standard freezer and up to 3 months at refrigerator temperatures, which simplifies transport, storage, and use in a pandemic [74].

According to a trial involving 44,325 people, this vaccine demonstrated 66.9% efficacy against symptomatic SARS-CoV-2 14 days after administration (95% CI, 59.0–73.4%), with higher efficacy in preventing severe-critical COVID-19, reaching 76.7% (95% CI, 54.6–89.1%) and 84.5% (95% CI, 54.2–96.9%) efficacy after 14 and 28 days after administration, respectively [73].

Concerning vaccine safety, injection-site pain was the most common local reaction (reported in 48.6% of the participants), while the most common systemic reactions were headache, fatigue, myalgia, and nausea. Some serious adverse events were considered by the investigators to be related to vaccination in the Ad26.COV2.S group, but a causal relationship between these events and Ad26.COV2.S could not be determined [73]. This vaccine induced high titers of neutralizing antibodies 2 and 4 weeks after injection, with the titer of neutralizing antibody significantly related to the level of protection. More than 80% of elderly and young subjects were positive for Th1 cytokines producing an S-specific CD4^+^ T-cell response with a very low or absent Th2 response, confirming the safety of this vaccine [75].

## 5. SARS-CoV-2 Vaccines and ADE

The main goal of all vaccines is to provoke an immune response against an antigen in the host and to have an effective and rapid response in case of a subsequent encounter with the antigen. A concern for SARS-CoV-2 vaccine development was that the immune response can enhance the disease, through the clearing of diseases. Understanding vaccine-induced immunopathology is important for all emerging infectious diseases because the mechanism of antibody-dependent enhancement (ADE) makes vaccine development particularly difficult due to its similarity to a natural infection [76].

Previous evidence for vaccine-induced ADE in animal models of SARS-CoV is conflicting and raises potential safety concerns. Liu et al., found that, while macaques immunized with a modified Ankara viral vector vaccine expressing the SARS-CoV S protein had reduced viral replication, anti-S IgG enhanced pulmonary infiltration of inflammatory macrophages, resulting in more severe lung injury compared with the unvaccinated animals [77]. In contrast, Qin et al., showed that an inactivated SARS-CoV vaccine protected cynomolgus macaques from viral challenge and did not result in enhanced lung immunopathology, even in macaques with low neutralizing antibody titers [78].

Preclinical studies of SARS-CoV immunization in animals led to results that vary greatly in terms of protective efficacy, immunopathology, and potential ADE, depending on the vaccine strategy employed.

Animal models were used to test SARS-CoV-2 vaccines specifically as the candidates were being assessed for ADE. ADE was evaluated and tested in vitro by Laczko et al., who designed nucleoside-modified mRNA vaccines encoding the full-length SARS-CoV-2 S protein encapsulated with LNPs (mRNA-LNP) that induced the production of high levels of S-protein-specific IgG. HEK293T cells expressing mouse FcgR1 were used to investigate the possibility of mRNA vaccines eliciting ADE, resulting in no SARS-CoV-2-associated ADE by testing the mRNA-vaccinated mouse sera [30]. In fact, mice that were given an mRNA vaccine expressing the prefusion SARS-CoV-2 S protein developed neutralizing antibodies and an S-protein-specific CD8^+^ response that was protective against lung infection without evidence of immunopathology [79]. Inoculation of ChAdOx1-S in rhesus macaques showed no pneumonia infection in vaccinated monkeys and no ADE effect [80].

ADE was monitored in human trials and at the start of the large-scale immunization campaign, and the results showed a lack of ADE-related events during the mass vaccination, which tentatively excludes the possibility that some vaccines might worsen the disease rather than improve or even prevent it [81].

## 6. Vaccine Delivery: Future Prospects

The mRNA and viral vector vaccines are delivered via the IM route, generating a systemic immune response lacking adequate levels of mucosal IgA and mucosal immunity. However, to neutralize pathogens at their initial entrance site, protective mucosal immunity is inevitably required. Therefore, an essential focus of vaccine development is related to their route of administration.

The respiratory mucosa plays a key role in SARS-CoV-2 infection and transmission; mucosal vaccination could thus represent an advanced approach to elicit simultaneous mucosal and systemic immune responses. Nevertheless, this alternative approach poses a greater challenge: the nucleic acid vaccine must penetrate the mucus layer, translocate into target cells, and evade extracellular and intracellular degradation. Mucosal immunization elicits an antibody response (specifically mucosal IgA antibodies), tissue-resident memory T-cells in the respiratory mucosa, and macrophage-mediated trained immunity, inhibiting further entry of pathogens [82]. Administration of mucosal vaccines has also been shown to elicit strong systemic humoral immunity, neutralizing any virus particle that evades the primary immune response at the mucosal site. For these reasons, immunization using an oral or intranasal vaccine could be an effective strategy for immune-prophylaxis against SARS-CoV-2 (Figure 2) [83]. This route of administration has advantages, not just in evoking a stronger immune response at both mucosal sites and systemic circulation, but also in offering the simplicity of needle-free administration, feasibility of mass vaccination without demanding expert medical staff, cost-effectiveness, and less severe side effects [84]. Viral vectors are one of the most promising strategies for mucosal vaccination due to their capacity for intracellular delivery, versatility, and intrinsic immunogenicity. An adenovirus type 5-vectored vaccine (AdCOVID) encoding the RBD of the SARS-CoV-2 S protein was tested in a single intranasal dose in inbred, outbred, and transgenic mice. Single intranasal vaccination elicited a strong, durable, and focused immune response against the RBD through the induction of mucosal IgA production in the respiratory tract, neutralizing antibodies in the serum, CD4^+^ cells with a Th1-like cytokine expression profile, and CD8^+^ cells. Moreover, a single intranasal dose completely protected mice from lethal SARS-CoV-2, preventing both weight loss and mortality. These data show that AdCOVID promoted concomitant systemic and mucosal immunity, making it a promising vaccine candidate [85].

Another vaccination strategy is the oral route, which presents the added challenges of low gastric pH and the difficulty in controlling where the nucleic acid payload is released. To address these challenges, nanocarriers and biomaterials have been used to develop innovative, protective delivery strategies for nucleic acids; polycationic materials, including chitosan and polyethylenimine (PEI), complexed with nucleic acids and encapsuled nucleic acid cargo by liposomes and polymersomes, are showing promising potential [86]. An oral tableted vaccine, the adenovirus-based VXA-CoV2-1/Vaxart, produced protection against SARS-CoV-2 in a Syrian hamster model in which the administration of two doses promoted a reduction in weight loss and lung pathology, and 5 days after injection, a complete deletion of the infectious virus was observed. Anti-spike IgG and neutralizing antibodies were induced upon oral immunization, with the serum demonstrating neutralizing activity [87]. Preliminary phase 1 trial results showed no severe adverse events and the triggering of multiple immune responses against SARS-CoV-2 antigens: a CD8^+^ cytotoxic T-cell response to the viral S protein that is necessary for long-lasting cross-reactive immunity, an increase in plasmablast cell numbers and an upregulation of the mucosal homing receptor, an increase in proinflammatory Th1 cytokines responsible for orchestrating the immune response to viral infection, and an IgA response in serum and/or nasal swab samples in 100% of the two-dose subjects [88,89].

## 7. How Variants Affect the Efficacy of SARS-CoV-2 Vaccines

The mutation rate of the virus should be considered in vaccine design, as genomic variations can induce immune evasion. For SARS-CoV-2, mutations in the S protein can induce conformational changes that may alter antigenicity, possibly affecting vaccine design and efficacy (Table 2) [90].

VOCs with increased transmissibility are contributing to the reversal of the decrease in COVID-19 cases that many countries have experienced.

The D614G variant first appeared in the United States in the middle of 2020. Since the second half of 2020, more SARS-CoV-2 variants have emerged, including B.1.1.7 (Alpha variant) first reported in the UK, B.1.351 (Beta variant) first reported in South Africa, P.1 (Gamma variant) first reported in Brazil, and B.1.617.2 (Delta variant) first reported in India, causing widespread concern [91].

Although B.1.1.7 variant has been de-escalated and is no longer circulating, it was considered a VOC because of its increased binding affinity to ACE2 and its enhanced ability to enter host cells, accelerating the spread of the virus. According to studies in the UK, the transmissibility of the B.1.1.7 variant reached up to 71% over and above the previous circulating strains of SARS-CoV-2 [92,93]. Analysis of individual-level data revealed an increase in COVID-19 mortality associated with the B.1.1.7 lineage [94,95], but no evidence was found regarding the ability of the B.1.1.7 variant to escape immunity [96].

B.1.351 is classified as a VOC because it increases the transmissibility and severity of the virus, but above all this variant appears less easily neutralized by convalescent plasma obtained from patients infected with previous variants and by serum obtained from vaccinees than the prototype virus on which vaccine antigens are based [94,97]. E484K mutation of the S protein has been reported to be related to the escape of the B.1.351 variant from neutralizing antibodies [98].

P.1 VOC shows increased transmission, risk of hospitalization, and mortality. Like the B.1.351 variant, P.1 was suspected to evade immunity in people who were previously infected [99]. A Centre for Arbovirus Discovery, Diagnosis, Genomics and Epidemiology (CADDE) study indicated variant P.1 to be twice as transmissible and was shown to be capable of evading ~32% of inherited immunity from previous coronavirus diseases, leading to the possibility of reinfection. These increased statistics were also reflected in fatalities, such that P.1 infections could be ~50% more lethal [100].

The B.1.617.2 variant was first identified in the Indian state of Maharashtra in late 2020 and spread throughout the country, outcompeting pre-existing lineages. This variant is highly transmissible, either due to higher viral burden or higher particle infectivity, resulting in a higher probability of person-to-person transmission. Interestingly this variant was found mainly in the younger population and the risk of COVID-19 hospital admission was approximately doubled in those with the B.1.617.2 when compared with the B.1.1.7 VOC, with risk of admission particularly increased in those with more relevant comorbidities [101]. In vitro, B.1.617.2 is six-fold less sensitive to serum neutralizing antibodies from recovered individuals. The first sub-lineage to be detected was B.1.617.1, followed by B.1.617.2, both bearing the L452R spike receptor binding motif mutation that was previously reported to confer increased infectivity and a modest loss of susceptibility to neutralizing antibodies [102].

A new variant of SARS-CoV-2, B.1.1.529 or Omicron was first reported to the WHO in South Africa on 24 November 2021. In recent months, there has been a steep rise in infections that coincided with the detection of the B.1.1.529 variant. On 26 November 2021, this variant was classified as a VOC [103].

The B.1.1.529 variant has a significant number of mutations, some of which are concerning: the spike gene contains nine mutations previously identified in other dangerous variants (Alpha, Beta, Gamma, and Delta), and three novel alterations not found in other VOCs. Almost nothing is known about the other eleven mutations, so it is not possible to evaluate whether these mutations are advantageous or disadvantageous regarding viral impact. Four mutations in the spike sequence modify the binding site of a monoclonal antibody and two mutations increase the affinity of the spike for human ACE2 receptors. Several other modifications appear to change the structure of parts of the spike, but their significance is still unknown. In addition to these mutations, several others have been found in other proteins of the virus [104,105].

The Omicron variant is the most divergent variant detected thus far. This has led to obvious concerns about its potential association with increased transmissibility, reduction in vaccine effectiveness, and an increased risk of reinfections [106]. Preliminary data indicate increased transmissibility and risk of reinfection, with the number of cases caused by the Omicron variant increasing in almost all South African provinces. Studies from South Africa and the UK reported doubling times of 3.38 days and 2–2.5 days, respectively, with the basic reproduction number (R0) above 3 and an estimated weekly increase in the Omicron-to-Delta ratio in the range of 7.2–10.2, which was considerably higher than the increase in the ratio of Delta to Alpha (estimated to be in in the range 2.5–4.2) [107,108]. However, the current limited and preliminary evidence suggests that Omicron has a less severe clinical presentation, presenting as a mild disease with predominantly upper respiratory symptoms such as muscle aches and fatigue for 1 to 2 days, headache, itchy throat, slight cold, or mild cough [109,110]. The UK Health Security Agency reported that Omicron-infected individuals have a 50% lower risk of visiting or to being admitted to the hospital than individuals infected with the Delta variant [111].

Regarding vaccine efficacy, Omicron demonstrates a greater breakthrough against vaccine-induced immunity as compared with Delta; however, vaccines should continue to prevent severe disease or death, although additional booster doses may be needed to protect most people from infection and to maintain the shield effect of immunizations [103,112,113].

Due to the presence of these variants, it is necessary to evaluate existing vaccines to determine if they are decreasing in efficacy against the variants and to decide whether modified or new vaccines are needed to restore efficacy against these variants.

### 7.1. BNT162b2

Low or no significant impact on vaccine effectiveness against the B.1.1.7 variant has been reported. A slight effect on serum neutralizing antibody titers from BNT162b2-immunized individuals was observed with a pseudovirus that contained the complete set of B.1.1.7 mutations, and even though there was a slight reduction, the Pfizer-BioNTech vaccine was found to be more than 90% effective against symptomatic infection from the Alpha variant [96]. The Pfizer-BioNTech vaccine serum neutralization activity showed a significant decrease against the B.1.351 variant (with an efficacy of 85%) compared with its activity against the B.1.1.7 variant, likely due K417N and E484K mutations [114]. Considering the high number of S protein mutations accumulated by P.1 variant, the B.1.1.7 variant was supposed to be equally or more resistant than the B.1.351 variant to antibody-mediated protection. However, laboratory serum neutralization assays using a pseudovirus have shown that the neutralizing activity of BNT162b2-elicited antibodies was roughly equivalent to the activity against the B.1.1.7 and P.1 virus [115]. A significant decrease in neutralizing antibody titer was observed for B.1.617.2 compared with B.1.1.7, using sera from individuals immunized with BNT162b2; furthermore, a vaccine effectiveness of 88% against symptomatic disease following infection with B.1.617.2 was observed after two doses of BNT162b2 [116].

Regarding the B.1.1.529 variant, recent data from South Africa have shown a 70% vaccine effectiveness against hospitalization during the Omicron-dominant period compared with the 93% effectiveness observed in the pre-Omicron period [113]. In vitro studies showed that, after two doses of BNT162b2, the serum had >22-fold reduced neutralizing titers against Omicron compared with a Wuhan pseudovirus. [112]. Live-virus neutralizing titers against Omicron after two doses of BNT162b2 showed a 29.8-fold reduction in the serum [117].

Despite this decrease in neutralizing titers, the BNT162b2 vaccine seems to maintain effectiveness against hospital admission [111]; furthermore, the addition of a booster dose of the vaccine may mitigate this reduction in vaccine effectiveness.

### 7.2. mRNA-1273

The neutralization capacity of the Moderna SARS-CoV-2 mRNA (mRNA-1273) vaccine against variants decreased similarly to that of the Pfizer-BioNTech vaccine. There was no significant impact on the neutralizing capacity of sera against the B.1.1.7 variant, with a reported 91% efficacy against the variant. Reduced levels of neutralizing antibodies against the other variants were reported, with 85% efficacy against P.1 and B.1.351 variants, and 70% efficacy against B.1.617.2 [114,116,118,119]. As with the BNT162b2 vaccine, recipients of the mRNA-1273 vaccine showed a decreased antibody response and memory B-cell response against SARS-CoV-2 variants containing E484K and N501Y mutations or the triple combination of K417N, E484K, and N501Y mutations [120]. There are limited data regarding vaccine efficacy against the Omicron variant; two preliminary studies found that two-doses of mRNA-1273 provided an initial effectiveness against omicron infection of 36.7% and 42.8%, which decline rapidly [121,122].

### 7.3. ChAdOx1

Regarding the ChAdOx1 vaccine, the laboratory virus-neutralizing activity of vaccine-induced antibodies was lower against the Alpha variant, demonstrating a vaccine efficacy of 70.4% [123]. A vaccine efficacy of 67.0% was reported against the B.1.617.2 variant [124] and 77.9% against the P.1 variant [125]. In a multicenter, double-blind, randomized-controlled trial, it was reported that the NAb titer against the B.1.351 strain was significantly reduced, with an efficacy of 10.4% [126], suggesting that this vaccine does not confer protection against mild-to-moderate COVID-19 caused by the Beta variant. An in vitro study showed a 36-fold reduction in neutralizing antibodies against the B.1.1.529 variant compared with the Delta variant [117]. An Imperial College report [127] showed a 5% vaccine effectiveness against the Omicron variant that did not confer protection against symptomatic infection, but still reduced hospitalization.

### 7.4. Ad26.COV2.S

Ad26.COV2.S vaccinated serum neutralized the B.1.1.7 variant in vitro, although less efficiently than the reference strain. The Johnson & Johnson vaccine appears to be effective against the B.1.617.2 variant 14 days after administration; the single-dose COVID-19 vaccine elicited neutralizing antibody production against the Delta variant at an even higher level than what was observed against the Beta (B.1.351) variant (vaccine efficacy of 52% against moderate disease) [73]. In one study, it was observed that multiple plasma specimens from recipients of the Ad26.COV2.S vaccine obtained 1 or 5 months after vaccination lacked detectable neutralizing activity against the Omicron variant [128]. One study observed that vaccine effectiveness against the B.1.1.529 variant for hospitalization was 63% [129].

In summary, a reduction in serum neutralization activity in vitro was observed against all VOCs, and there was evidence of infection with VOCs in vaccinated populations. However, the protection efficacy is still higher than the protection level criterion of at least 50%, as defined by the WHO; mostly importantly, the severity of disease is still reduced following vaccination, indicating that these vaccines are still highly effective. Other VOCs will surely emerge, and the impact of these variants on vaccine efficacy is difficult to predict.

Currently, novel vaccination strategies are being explored to increase vaccine efficacy against variants; increasing the number of doses and heterologous vaccination to boost the immune response would help to prevent further transmission of variants, raising the vaccine-induced immune response level.

Moreover, the next generation of vaccines, such as multivalent vaccines, would be effective tools to control the spread of SARS-CoV-2 variants.

## 8. Recommendations on Extra Doses and Boosters

Despite the initially promising results of vaccination campaigns, many countries are experiencing a resurgence of COVID-19 dominated by the Delta (B.1.617.2) and Omicron (B.1.1.529) variants. Two important factors challenge the success of the vaccination campaigns against COVID-19: first, the emergence of new variants that may evade immune responses in previously vaccinated subjects; and second, the antibody titers may significantly decrease over time [130].

A study conducted by the Italian Istituto Superiore di Sanità (ISS) confirmed the increased efficacy in preventing cases of COVID-19 and severe disease (hospitalization and/or hospitalization in intensive care and/or death) in individuals who had been fully vaccinated for more than 6 months or less compared with the unvaccinated. Moreover, it was stated that, 6 months after the completion of the vaccination cycle, there was a significant decrease in vaccine efficacy in preventing diagnose in all age groups. In general, over the entire fully vaccinated population, vaccine efficacy starts at 76% and decreases to 50% over 6 months. In cases of severe disease, there is a small decrease in vaccine efficacy: the efficacy decreases from 92% to 82% in fully vaccinated individuals after 6 months [131].

Due to increased COVID-19-related hospital admission, the Israeli Ministry of Health started a campaign for administration of a third dose of the BNT162b2 mRNA COVID-19 vaccine based on evidence reporting a pronounced humoral response to a third dose of mRNA vaccine [132,133].

In Israel, the administration of a booster dose of the BNT162b2 vaccine was first approved to immunocompromised patients and then expanded to people 60 years of age or older who had received a second dose of vaccine at least 5 months earlier [134,135]. Following an observational analysis conducted on this population [134], a third dose of the BNT162b2 mRNA COVID-19 vaccine was reported to be effective in preventing severe COVID-19-related outcomes. Vaccine effectiveness was evaluated at least 7 days after receipt of the third dose 5 months after the second dose and compared with receiving only two doses: the efficacy was estimated to be 93% (231 events for two doses vs. 29 events for three doses) for hospital admission, 92% (157 vs. 17 events) for severe disease, and 81% (44 vs. 7 events) for COVID-19-related death [134,135].

Based on the Israeli campaign data, it was established that a third booster dose of two-dose Comirnaty vaccine was needed to restore vaccine efficacy, distinguishing between the booster dose for individuals with weakened immune systems and for individuals with normal immune systems.

On 4 October 2021, the EMA recommended boosters of COVID-19 vaccines based on studies reporting that an extra dose of these vaccines increased the production of antibodies against SARS-CoV-2 in organ transplant patients with weakened immune systems [136,137].

At first, the EMA Human Medicines Committee (CHMP) recommended an extra dose of the Comirnaty and Spikevax vaccines for people with severely weakened immune systems, administered at least 28 days after their second dose [138].

The EMA CHMP also evaluated data on antibody levels after a booster dose of Comirnaty and Spikevax vaccines administered to people with normal immune systems approximately 6 months after the second dose, and the results showed increased antibody levels. On the basis of this data, the CHMP concluded that booster doses may be considered at least 6 months after the second dose for people aged 18 years and older with normal immune systems [138,139].

The EMA CHMP has also concluded that a booster dose of the Janssen vaccine may be considered at least 2 months after the first dose in people aged 18 years and older [140]. The FDA also approved a booster dose of the Comirnaty and Spikevax vaccines for individuals aged 18 years and older at least 6 months after completion of the primary vaccine series [141]. A single booster dose of the Janssen vaccine was also approved for individuals aged 18 years and older, to be administered at least 2 months after completion of the primary single-dose regimen [142].

Triple-dose vaccinations series have been initiated in many countries and their data will be helpful in characterizing the potentially boosted immunity and therefore the clinical importance of the triple-dosing of COVID-19 vaccines [130].

A retrospective study was conducted in a large Israeli health maintenance organization, and it was found that the infection rate of COVID-19 in the booster-vaccinated population decreased by 3.8% compared with the population who did not receive a booster dose [143]. Due to the spread of the SARS-CoV-2 Omicron variant in several countries, the administration of a third dose has been accelerated, driven by the concern about lower vaccine efficacy with this variant.

Data from the UK Health Security Agency on hospital admissions after Omicron infection and vaccine effectiveness analysis reported 72% protection after two doses of the BNT162b2 vaccine for up to 6 months, rising to 88% within 2 weeks of receiving a booster dose [111].

A third dose of the mRNA-1273 vaccine showed a vaccine efficacy of 67.7% (compared with the ~40% efficacy of two doses) against Omicron infection [121]. Data from a South Africa study involving healthcare workers (*n* = 227,310) who received the single-dose Johnson & Johnson COVID-19 vaccine as a primary dose showed that the booster increased vaccine effectiveness against hospitalization to 85% [129]. From these early data, it can be deduced that, with a third dose, the antibodies generated are able to effectively neutralize the virus, bringing protection from infection and serious disease back to high levels. It was also observed that a third dose could substantially reduce transmission, especially in highly vaccinated populations. A modeling study by Gardner et al., showed that titers of neutralizing antibodies are strongly correlated not just with protection against symptomatic disease, but also with protection against all infection and transmission [144].

On 3 January 2022, Israel began to offer a fourth dose of the COVID-19 vaccine to healthcare workers and adults over the age of 60 years. Fourth doses are mainly administered on a precautionary basis to control the spread of the Omicron variant. To date, there is very little evidence for their effectiveness, either from studies of immune function or from observational studies of clinical events. The prime minister of Israel, Naftali Bennett, stated that in a small unpublished study conducted on 154 hospital employees, there was a five-fold increase in antibody concentrations in subjects one week after receiving a fourth dose of the Pfizer-BioNTech vaccine, likely leading to increased protection against severe symptoms and hospital admission [145,146].

## 9. A Proven “old” Strategy as a New Weapon against COVID-19

As described above, several vaccines using different platforms are currently available; however, it is important to have many vaccines options using different technologies to meet the various needs of individuals with pre-existing conditions and to increase global production to satisfy the needs of low- and middle-income countries. On 20 December 2021, the EMA recommended granting conditional marketing authorization for the Novavax COVID-19 vaccine Nuvaxovid (also known as NVX-CoV2373) [147]. This vaccine was engineered from the genetic sequence of the first strain of SARS-CoV-2 and was formulated using Novavax’s recombinant nanoparticle technology to produce antigens derived from the coronavirus S protein; the antigens were combined with an adjuvant to enhance the immune response and the production of neutralizing antibodies patented by Novavax [148]. A trial in the UK reported the efficacy of this protein-based vaccine as 96.4% against the original virus strain, 86.3% against the Alpha (B.1.1.7) variant, and 89.7% overall [149]. Protein-based vaccines have been used for decades against hepatitis, shingles, and other viral infections, and they have several advantages as they are usually safe, well tolerated, and do not trigger autoimmunity. It must be noted that these approaches are also favorable in terms of cost, simplicity, production capacity, transport, and administration, with fewer standardized steps for their production and manufacturing; these attributes make protein-based vaccines ideal candidates for global production and usage, making them affordable and more available all around the world for both developed and developing countries [150].

## 10. Conclusions and Perspectives

COVID-19 is an unusual global health threat where a vaccine was needed immediately as vaccine immunization is, thus far, the best solution in preventing the spread of infectious disease. Exceptional efforts have been made to develop effective and safe vaccines to be released rapidly at an affordable cost to all countries affected by the pandemic [37]. In spite of this, much remains to be understood and improved upon regarding the COVID-19 pandemic.

As previously stated, an essential focus of study for future vaccines is their route of administration. All mRNA and viral vector COVID-19 vaccines are currently delivered via the IM route and generate a systemic immune response. Nevertheless, some authors have explored the potential role of vaccines able to induce immunological responses directly in the respiratory mucosa, similar to those developed for influenza or measles, which would likely be more effective in the early control or clearance of SARS-CoV-2 [84].

Whether long-lasting protection can be provided by COVID-19 vaccines still remains unclear, and it is essential to determine how long a protective immune response against SARS-CoV-2 infection can be maintained in individuals. [151]. Moreover, because reinfection has been reported in vaccinated individuals and vaccine effectiveness against asymptomatic infection and mild COVID-19 disease may decrease over time and with the emergence of new variants, a third booster dose of Comirnaty, Spikevax, and Janssen vaccines has already been approved by the EMA and FDA [139,142,143].

The emergence of novel SARS-CoV-2 strains is a central consideration for present and future vaccines. In this regard, a promising objective could be to achieve the development of pan-coronavirus vaccines targeting the entire *Coronaviridae* family, eliciting cross-reactive antibodies that may be more effective against variants of a single virus compared with strain-specific antibodies [152], although this concept is still in its infancy.

Efforts to tackle the COVID-19 pandemic could be thwarted by the unequal distribution of SARS-CoV-2 vaccines between high-income and low-income countries. The poor supply of COVID-19 vaccines to low-income countries could extend the duration of the pandemic and could favor the emergence of novel SARS-CoV-2 variants [153]. Furthermore, this unfair distribution, highlighted by the initiation of fourth-dose administration in some countries, poses a serious ethical problem for the political leaders of high-income countries [153,154]. The WHO director-general, Tedros Adhanom Ghebreyesus, has defined the unequal distribution of COVID-19 vaccines as a “catastrophic moral failure” and as “another brick in the wall for the inequality between the world’s haves and have-nots” [155].

Therefore, the current potential for variants to escape vaccine-induced immunity makes testing and sequencing a priority to rapidly identify new variants, initiate a third booster dose, develop next-generation vaccines that elicit broad neutralizing activity, attain an equal distribution of COVID-19 vaccines among high-income and low-income countries, and, last but not least, to increase people’s trust in scientific evidence.

## Figures and Tables

**Figure 1 vaccines-10-00349-f001:**
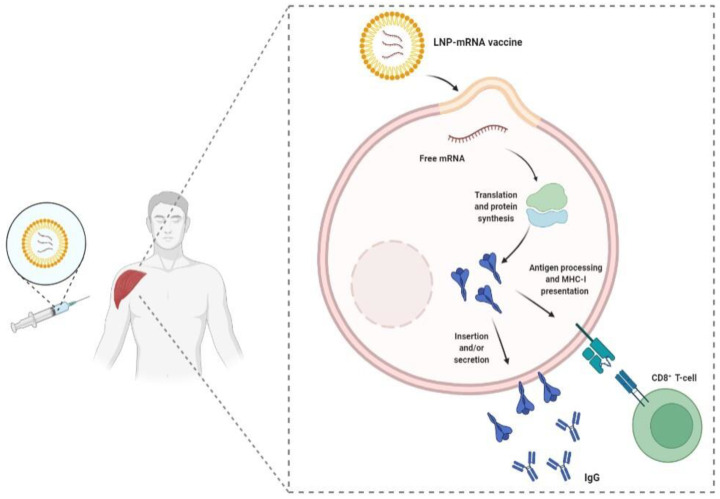
Simplified overview of the mRNA vaccine mechanism of action. Once the mRNA molecule is released from the LNP into the cytosol, it is directly translated by ribosomes into polypeptides. Polypeptides can be processed by the proteasome system, leading to peptide presentation to CD8^+^ T-cells via MHC I and can also undergo post-translational modifications and be folded into a protein that can either be membrane anchored or secreted to induce antibody production.

**Figure 2 vaccines-10-00349-f002:**
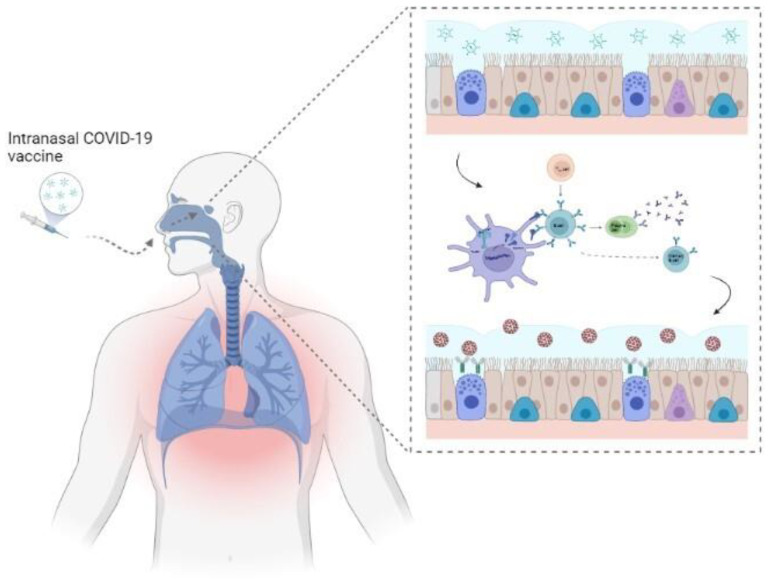
Principles of mucosal vaccination. Mucosal vaccine administration stimulates the synthesis of immunoglobulin A (IgA) at the level of mucosa-associated lymphoid tissue (MALT). The mucosal immune response is characterized at the local level by secretory IgA (SIgA) and cytotoxic T-cells, and at the systemic level by antigen-specific humoral and cellular responses. These outcomes have been shown to be effective in clearing various pathogens, including respiratory viruses.

**Table 1 vaccines-10-00349-t001:** SARS-CoV-2 EMA/FDA-authorized vaccines.

Features	BNT162b2	mRNA-1273	ChAdOx1	Ad26.COV2.S
Vaccine Type	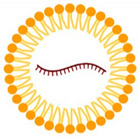 mRNA		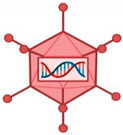 Viral Vector	
Manufacturer	Pfizer-BioNTech (US—GER)	Moderna (US)	AstraZeneca/Oxford (UK)	Janssen Pharmaceutical/Johnson & Johnson (US)
Commercial Name	Comirnaty	Spikevax	Vaxzevria	Janssen COVID-19 Vaccine
Antigen	Full-length spike (S) protein with proline substitutions	Full-length spike (S) protein with proline substitutions	Replication-deficient chimpanzee adenoviral vector with the SARS-CoV-2 S protein	Replication- deficient human adenovirus serotype 26 vector encoding a full-length, stabilized SARS-CoV-2 S protein
Dose	30 μg	05 mL	5 × 10^10^ Viral particles	5 × 10^10^ Viral particles
Dosage	2 Dosed 21 d apart	2 Dosed 28 d apart	2 Dosed 28 d apart	1 Dose
Storage Condition	−80 to −60 °C; 2–8 °C for 5 d; RT ≤ 2 h	−25 to −15 °C; 2–8 °C for 30 d; RT ≤ 12 h	2–8 °C for 6 months	20 °C; 2–8 °C for 3 months
Efficacy	94.6% 7 d after 2 doses	94.1% 14 d after second dose	70.4% 14 d after second dose	66.9% 14 d after administration
Serious Adverse Event	Anaphylaxis and myocarditis	Myocarditis, anaphylaxis, and other serious allergic reactions	Cerebral venous sinus thrombosis and other venous thrombosis	Cerebral venous sinus thrombosis and other venous thrombosis

**Table 2 vaccines-10-00349-t002:** Effect of variants on vaccine efficacy.

SARS-CoV-2 Strains	BNT162b2	mRNA-1273	ChAdOx1	Ad26.COV2.S
SARS-CoV-2 Wild-Type strain	94.6% 7 d after 2 doses	94.1% 14 d after second dose	70.4% 14 d after second dose	66.9% 14 d after administration
B.1.1.7 Variant (Alpha)	90% (85–95%)	91% (84–95%)	70.4%	Effective
B.1.351 Variant (Beta)	85% (70–93%)	85% (80–90%)	10,4% (−77%–55%)	52%
P.1 Variant (Gamma)	88% (75–92%)	85% (80–90%)	78% (69–84%)	Effective
B.1.617.2 Variant (Delta)	88% (75–90%)	70% (45–85%)	67% (61–72%)	Effective
B.1.1.529 Variant (Omicron)	29.8 fold decrease	36.7–42.8%	5%	63%
References	[96,114,115,116,117]	[114,116,118,119,121,122]	[123,124,125,126,127]	[74,129]

## Data Availability

Not applicable.

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
