# Peer review of "COVID-19 Vaccine: Between Myth and Truth"

_vaccines, 2022, doi:10.3390/vaccines10030349_

Round 1

Reviewer 1 Report

Many excellent reviews have already been published on this subject, several with the same structure 10.1038/s41577-021-00592-1, 10.3390/ph14050406. The manuscript does not provide a new perspective or a novel organisation of the data. I am afraid that all interested readers in this competitive field have already found and read this information.

The use of references is inaccurate, which significantly reduces the credibility of the article for the reviewer.

Due to some misspellings and inaccuracies, the manuscript needs to be re-edited

The authors cite reference 17 as the source of the sentence:: „Clinical trials in humans demonstrate that free mRNA released from the lipid nanoparticle are recognized by toll-like receptor 3 (TLR3), TLR7, TLR8, or retinoic acid-inducible gene (RIG-I), that activates IFN-1 production and stimulates the Th1 response

In contrast, in the article cited, we read: „Furthermore, they were able to show that mRNA molecules possessing this modification did not trigger pathogen-associated molecular pattern sensing mechanisms such as toll-like receptors (TLRs) or retinoic acid-inducible gene I (RIG-I)” 

New results are available on the adjuvant effect of RNA vaccines:  10.1038/s41578-021-00358-0,   10.1016/j.immuni.2021.11.001

The authors cite reference 107 as the source of the sentence:: „Felten et. al [107] even patients treated with rituximab able to develop anti-SARS-CoV-2 humoral response after two vaccine doses, acquire a neutralizing antibody response or to develop a T-cell response, or both after a third dose

This link must refer to another article by the same author, the correct one is 10.1016/S2665-9913(21)00351-9.. The interpretation of this article by the authors may lead to misunderstandings. According to the original article „Median time between last rituximab infusion and first dose of vaccine was 227 days” and even after this period, only a few patients had antibody levels.

This article should also be cited here 10.1038/s41591-021-01507-2

Other references (for ex: 86-89) are from non-scientific sources (New York times, South African physicians…) and should not be cited in a peer-reviewed article.

Author Response

Dear reviewers,

Many thanks for your comments and suggestions.

Since an improvement of the language was highly recommended, we submitted the manuscript for proof-reading of English by native English speaking academic editor from Cambridge Proofreading LLC as we reported at the end of the manuscript.

We have also carefully checked the references correspondence, which is now correct. We apologize for the inconvenience in first submission. By switching from Mac to PC the reference manager software didn’t work proèerly.

Reviewer #1:

  • “Many excellent reviews have already been published on this subject, several with the same structure 10.1038/s41577-021-00592-1, 10.3390/ph14050406. The manuscript does not provide a new perspective or a novel organisation of the data. I am afraid that all interested readers in this competitive field have already found and read this information.”

We have added some novelties regarding the skeptiscism toward vaccine and in particular COVID-19 vaccine. Moreover, since in our review we do describe Omicron variant effects on vaccine efficacy and the effect of the booster dose, we think this may represent a novelty that could capture the interest of readers.

  • We have corrected the meaning of the sentence (line 217): “Clinical trials in humans demonstrate that free mRNA released from the lipid nanoparticle are recognized by toll-like receptor 3(TLR3), TLR7, TLR8, or retinoic acid-inducible gene (RIG-I), that activates IFN-1 production and stimulates the Th1 response” referring to the paper by Bettini et al. (Vaccines 2021; ref 35).

  • We added the new results suggested for adjuvant effect of RNA vaccines (lines 202-207; ref 30, 32).

  • Regarding the citation of reference 107, we decided to cut off this part from our manuscript since it was referring to the immunocompromised individuals which were not the proper target of our review.

We have removed the non-scientific articles we have cited before.  

Reviewer 2 Report

Figure 1: Explain and illustrate with more details Section 10.2 & 10.3 : Describe in-depth with recent study

Table 2: Add on the reference Colum Conclusion must short and information.

Rephrase the the conclusion and arrange the data from this section to other suitable section

It will be preferable to add figure for more interest to the reader Is it CD4+ Th1 cells, CD8+ T-cell or "+" will be superscript form.

Verify please 270.031.622 & 5.310.502 ? It is 270,031,622 & 5,310,502!

Through out the manuscript I have found same issue.

CD8 and CD4 T helper ?

Verify please "This vaccine demonstrated a 94.1% efficacy (95% CI, 89.3%-96.8%), in the trial (NCT04470427) involving around 30,000 people in total" rephrase the sentence

Author Response

Dear reviewers,

Many thanks for your comments and suggestions.

Since an improvement of the language was highly recommended, we submitted the manuscript for proof-reading of English by native English speaking academic editor from Cambridge Proofreading LLC as we reported at the end of the manuscript.

We have also carefully checked the references correspondence, which is now correct. We apologize for the inconvenience in first submission. By switching from Mac to PC the reference manager software didn’t work properly.

Reviewer #2

  • We have explained with more details Figure 1 (which is now become Figure 2).

  • We have described in-depth section 10.2 and 10.3 (which are now 7.2 and 7.3) with recent studies.

  • We have added “reference” column in Table 2.

  • We have rephrased the conclusion section to make it shorter and more concise. We have arranged the data that were in this section in a separated section (section 9).

  • We have added Figure 1 (“Simplified overview of the mRNA vaccine mechanism of action.”) to be more interesting for the reader.

  • We have changed the form of CD8+ T-cells by making it superscript.

  • We have replaced the “.” with the “,” in the numbers through the manuscript.

  • We have not understood what you meant with the suggestion “CD8 and CD4 T helper?”.

  • We have verified and rephrased the sentence “This vaccine demonstrated a 94.1% efficacy…” (line 338).

Reviewer 3 Report

Dear Authors,

It was my pleasure to review the manuscript which you submitted, titled COVID-19 Vaccine: between Myth and Truth 2, a scoping review dealing with a most consequential and vital issue facing humanity right now: the struggle against the SARS-CoV-2 pandemic and the best weapon curently available to put an end to it and restore normalcy, public health and economic development.
I believe the authors have thoroughly and competently laid out their overview, making their contribution valuable for both researchers and lay people seeking to learn more about the many complexities which the current pandemic scenario entails. Hence, I would give the authors credit for successfully expounding upon pandemic and vaccination dynamics in a rather comprehensive fashion.
The article's title says "myth and truth", yet I do not see any "myths" addressed by the authors. They should at least in passing address some of those "myths" based on misinformation spread by vaccine skeptics and conspiracy theorists that frequently hinder vaccination efforts, fueling vaccination hesitancy and making things worse in terms of mortality and saturation of health care facilities.
By the same token the several caveats by the WHO and others, supported by recent data concerning booster shots should also be mentioned and commented on, especially in light of the fact that as some nations move into a fourth dose, developing countries have extremely low vaccination rates. That may entail epidemiological repercussions of major magnitude in terms of the developments of new variants which may undermine efforts to end the pandemic. Hence, while boosters are indeed valuable particularly for vulnerable individuals, their generalized widespread use use remains controversial. It has been argued that primary vaccination should be prioritized for as many peolpe at risk as possible, all over the world, rather than providing booster shots for those living in wealthy countries who are not as much at risk. Our ability to boost our way out of the pandemic is not to be taken for granted. Certainly, there is no silver bullet, and the policy adjustments in that regard may prove extremely complex to implement.

Overall, the article is well-written, but I would advise the authors to have it proof-read again by a native speaker of English. There are a few instances of clumsy grammar that need to be fixed, such as:

Still, it is but remain still important to have more as many vaccines using different as many technologies as possible, in order to meet different needs and of individuals with pre-existing conditions as well as to increase the global production in order and be able to satisfy the necessity of low- and middle-income countries (lines 615-618).

All in all, I believe that although there may be little in terms of novelty, this review is a valuable contribution and one worth publishing, provided that a few adjustments are made in terms of scope and readability. 

Sincerely,

Author Response

Dear reviewers,

Many thanks for your comments and suggestions.

Since an improvement of the language was highly recommended, we submitted the manuscript for proof-reading of English by native English speaking academic editor from Cambridge Proofreading LLC as we reported at the end of the manuscript.

We have also carefully checked the references correspondence, which is now correct. We apologize for the inconvenience in first submission. By switching from Mac to PC the reference manager software didn’t work properly.

Reviewer #3

  • We have added a section addressing “myths” (section 2).

  • We have discussed about the use of booster shots vs primary vaccination in the conclusion section (section 10).

We hope that the improvement we have made on the manuscript thanks to your suggestion, can meet your requirements.

We remain at your complete disposal to further clarification.

Your sincerely,

Round 2

Reviewer 1 Report

The novelty of the manuscript is minor, while a number of reviews have appeared in this field.

Reviewer 2 Report

Accept in present form